# Early changes in the properties of CA3 engram cells explored with a novel viral tool in mice

Dario Cupolillo[†], Noelle Grosjean, Catherine Marneffe, Julio Viotti, Celia Reynaud, Severine Deforges, Mario Carta, Christophe Mulle*

Interdisciplinary Institute for Neuroscience, CNRS UMR 5297, University of Bordeaux, Bordeaux, France

## eLife Assessment

This **important** study characterizes and validates a new activity marker - fast labelling of engram neurons (FLEN) - which is transiently active and driven by cFos, allowing the monitoring of intrinsic and synaptic properties of engram neurons shortly after the learning experience. The results **convincingly** demonstrate the utility of this novel viral tool for studying early changes in the properties of engram cells. FLEN will provide a beneficial tool for the neuroscience community once it is made available at a plasmid repository.

**\*For correspondence:**
christophe.mulle@u-bordeaux.fr

**Present address:** [†]Italian Institute of Technology, Synaptic Plasticity of Inhibitory Networks, Genova, Italy

**Competing interest:** The authors declare that no competing interests exist.

**Abstract** Forming new memories after a one-time experience requires initial encoding then consolidation over time. During learning, multimodal information converges onto the hippocampus, activating sparse neuronal assemblies which are thought to form a memory representation through concerted activity and synaptic interconnectivity. In this work, we use a novel tool for fast fluorescent labeling of engram neurons (FLEN). FLEN is based on c-Fos activity-dependent transient expression of a destabilized fluorescent marker ZsGreen1 rapidly after one-trial learning. With FLEN, we explore the electrophysiological properties of c-Fos activated CA3 pyramidal neurons (PNs) a few hours following one-trial learning of an episodic-like memory. In parallel, we employ the Robust Activity Marker (RAM) system, which provides activity-dependent labeling 24 hr following a novel experience. Comparing FLEN+ and RAM+ neurons allows us to characterize how the properties of neuronal assemblies evolve during an initial phase of consolidation. Whereas no difference was observed in the excitability of FLEN+ vs. FLEN− neurons, RAM+ neurons were more excitable than RAM− neurons. This suggests that CA3 PNs recruited in an engram progressively acquire increased excitability as compared to neurons which were not activated by the one-trial contextual memory task. On the other hand, like RAM+ neurons, FLEN+ CA3 neurons show an increased number of excitatory inputs. Overall, with the FLEN strategy, we can show that both the intrinsic excitability and the synaptic properties of CA3 PNs undergo progressive plastic changes over the first day following a one-trial memory task.

## Introduction

Animal survival depends on the ability to rapidly form new and durable episodic memories of one-time meaningful events which, when recalled, allow to adapt behavior (*Tulving, 2002*). Contextual memories of ongoing experiences are continuously and rapidly generated in the hippocampus and subsequently stored in an extended cortical network (*Eichenbaum, 2000*). Computational models of memory point to a crucial role of the hippocampal CA3 region in episodic memory encoding and

retrieval (*Kesner and Rolls, 2015*). Based on its distinctive extensive recurrent architecture (CA3–CA3), CA3 may act as an auto-associative or attractor network, enabling pattern completion during recollection (*Kesner and Rolls, 2015*; *McNaughton and Morris, 1987*). Indeed, mice with CA3-specific synaptic plasticity impairment (CA3 NR1-KO mice) are unable to encode the novel location of a hidden platform in a delayed matching-to-place version of the Morris water maze (*Nakazawa et al., 2003*) and to retrieve the learned position of a submerged platform when partial cues are presented (*Nakazawa et al., 2002*). In addition to recurrent connections, the main excitatory synaptic drivers of CA3 pyramidal neurons (PNs) are inputs from medial and lateral entorhinal cortex via the perforant pathway, and indirectly from mossy fibers (Mfs) through the dentate gyrus (DG). Mf inputs display 'conditional detonator' properties, enabling individual Mf synapses to trigger postsynaptic firing in response to repetitive firing of presynaptic DG cells (*Henze et al., 2002*; *Marneffe et al., 2025*; *Rebola et al., 2017*; *Sachidhanandam et al., 2009*; *Vandael and Jonas, 2024*).

During encoding of novel information, the orchestrated activity patterns of selected neuronal groups distributed throughout the brain, called engrams, represent the initial memory trace (*Josselyn and Tonegawa, 2020*). Engrams undergo long-lasting synaptic modifications during consolidation and are selectively re-engaged when the memory is retrieved (*Bessières et al., 2024*; *Goto et al., 2021*; *Josselyn and Tonegawa, 2020*; *Kitamura et al., 2017*; *Ryan et al., 2015*; *Tomé et al., 2024*). Experimental manipulation of neuronal engrams in the hippocampus allows selective memory erasure (*Han et al., 2009*), artificial memory recall (*Liu et al., 2012*), and creation of a synthetic memory (*Vetere et al., 2019*), highlighting both their necessity and sufficiency for memory functions.

The investigation of the properties of engram neurons has taken advantage of immediate early genes (IEGs) such as c-Fos, Arc (activity-regulated cytoskeleton-associated protein), or *Npas4* (neuronal PAS domain protein 4) to visualize and label neurons active during a memory task (*DeNardo et al., 2019*; *Kawashima et al., 2013*; *Reijmers et al., 2007*; *Sørensen et al., 2016*; *Sun et al., 2020*). Generally, IEGs are rapidly and transiently expressed; however, different IEGs are triggered by different sets of extracellular events and possess specific expression kinetics. Temporal control of IEG-based systems is crucial to accurately link engram labeling to specific events that occur in a specific time.

The activity of engram neurons can be tracked in vivo during their maturation from the process of encoding throughout consolidation in behaving mice, by means of functional indicators like GCaMP (*Milczarek et al., 2018*; *Mocle et al., 2024*; *Tomé et al., 2024*). However, in vivo Ca$^{2+}$ imaging does not allow addressing the synaptic and intrinsic properties of engram neurons. Labeling of engram neurons in order to characterize these properties ex vivo over time after an experience has occurred often relies on labeling techniques using inducible transgenic systems (*DeNardo et al., 2019*; *Reijmers et al., 2007*). In several studies using transgenic mouse lines, engram labeling requires 48 hr of doxycycline (Dox) deprivation before a novel experience (*Kitamura et al., 2017*; *Liu et al., 2012*; *Pignatelli et al., 2019*; *Ryan et al., 2015*), or the use of tamoxifen injection to open a 12-hr time window for engram labeling (*DeNardo et al., 2019*). Therefore, these systems only offer temporal control over extended time windows, thus increasing the probability of compromising the experience specificity of labeled engram cells. In addition, most of these approaches drive a relatively late labeling, enabling investigation of learning-induced modifications more than 24 hr after a novel experience (e.g. Robust Activity Marker [RAM] system) (*Sun et al., 2020*) or shortly after recall for neurons which have been labeled several days before (*Pignatelli et al., 2019*), when consolidation processes have already been engaged.

Little has been learned about how neuronal engrams in the hippocampus rapidly (in the time range of hours) adapt their synaptic and physiological properties following encoding of a memory event, and how these properties evolve over time. In this sense, it is unknown whether initial memory storage is enabled by engram cell modifications of intrinsic properties, synaptic plasticity, or a combination of them. Here, we tackle these questions using a combination of viral strategies that allow us to identify engram neurons and compare ex vivo their properties shortly after learning (3–4 hr), and 24 hr after the experience has occurred (*Figure 1A*; *Cupolillo, 2021*). To track early engrams, we developed a virally delivered c-Fos-based genetic construct that provides physiological-like rapid expression and fast decay of the bright fluorescent cytosolic marker ZsGreen1. This novel viral strategy for fast and efficient labeling of engram neurons (FLEN) allows identifying and characterizing engram neurons within 3–4 hr following a novel experience, in reference to non-activated neurons. Then, to understand how initial engram cell properties evolve within a 24-hr time scale, we employ the more durable

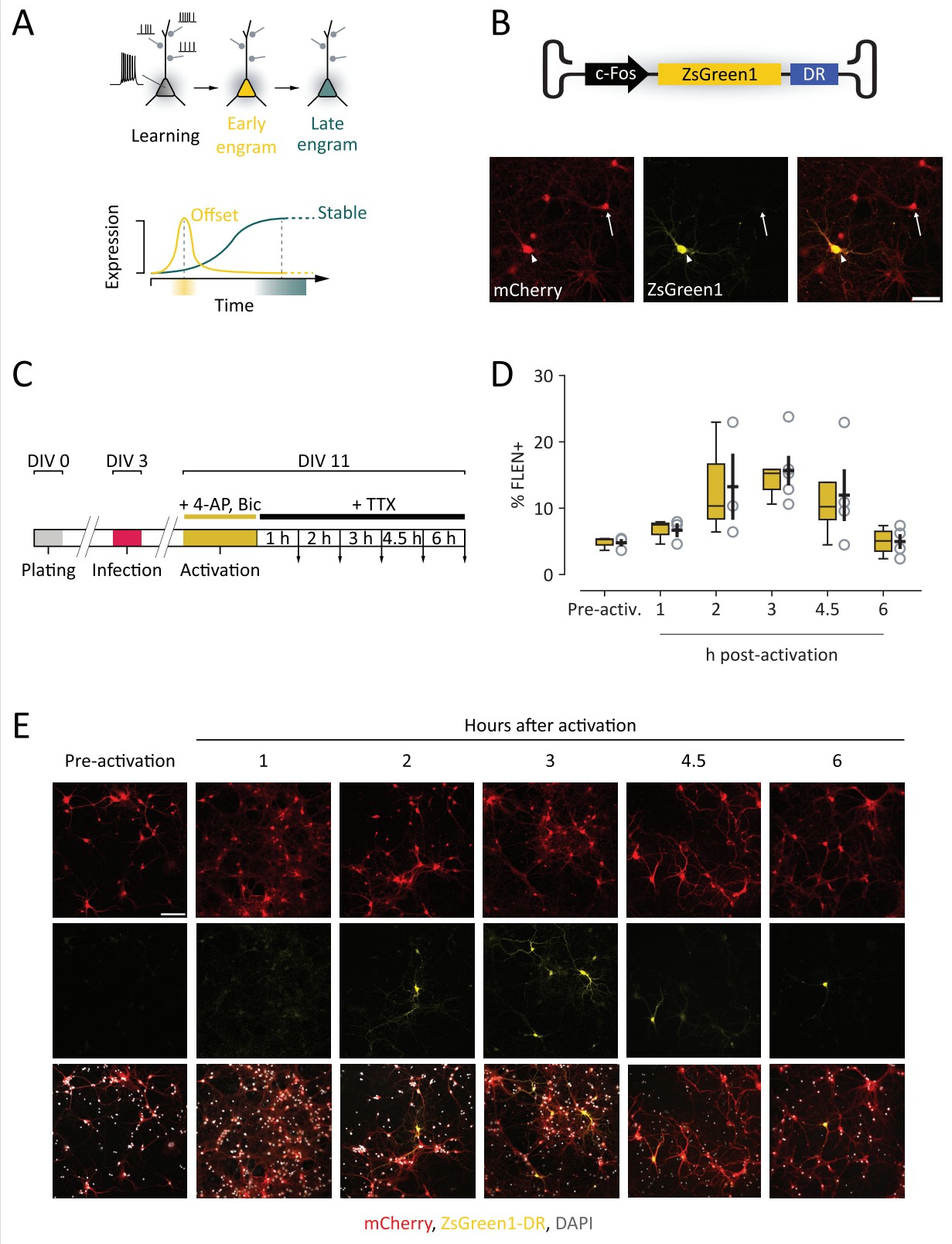

**Figure 1.** Development of a novel viral tool to study early changes in the properties of CA3 engram cells. (**A**) Scheme representing the maturation of engram neurons during memory formation. Top panel: during learning, a neuron receives numerous synaptic inputs, resulting in sequences of action potentials. This activity triggers the rapid expression of immediate early genes (IEGs), marking the neuron as part of an engram (yellow). As the engram matures (green), IEG expression is not detectable anymore, yet the neuron retains established engram properties. Bottom panel: expression function of

*Figure 1 continued on next page*

Figure 1 continued

IEG-based construct to investigate and compare early and late engram properties. A construct with rapid offset (yellow) best captures engrams shortly after learning, whereas a progressively expressing construct best captures more mature engrams. (**B**) FLEN construct (c-Fos.ZsGreen1-DR) encapsulated in AAV2/9 vector. Bottom panel: representative images showing ZsGreen1 fluorescence in c-Fos+ neurons (arrowhead) and c-Fos− neurons (arrow). Scale bar = 50 μm. (**C**) Timeline of in vitro experiments to determine FLEN expression and offset time course. After infection (DIV3) and activation (DIV11), samples are washed with a blocking medium and images are acquired at different intervals (downward arrows). (**D**) Percentage of ZsGreen+ vs. all transduced neurons (mCherry+ cells) at the different intervals following activation. (**E**) Representative images of activated cultured neurons at 1, 2, 3, 4.5, and 6 hr after activation. Scale bar = 100 μm.

c-Fos-based artificial RAM system (*Sørensen et al., 2016*) to label neurons and characterize their properties 24 hr after the exposure to a one-trial memory task.

## Results

### Creation and validation of c-Fos-ZsGreen1-DR for fast labeling of engram neurons (FLEN)

We generated a genetic cassette for adeno-associated virus-mediated delivery which allows fast labeling of engram neurons (FLEN). This construct includes the full-length, activity-dependent c-Fos promoter which controls the expression of a destabilized variant of the green fluorescent protein ZsGreen1 (*Matz et al., 1999*; *Figure 1B*). The complete sequence of the c-Fos promoter ensures a physiological-like gene expression of ZsGreen1, which follows c-Fos expression. ZsGreen1 provides a bright whole-cell somatic and dendritic labeling of activated neurons (*Figure 1B*). To restrict the labeling time window and minimize labeling unrelated to the designed experience, ZsGreen1 was fused with a destabilizing domain (DR); when fused to EGFP, DR decreases the fluorescence half-time to 2 hr (*Li et al., 1998*).

To determine the expression kinetics of FLEN in response to neuronal activity, P0 mouse cortical neurons were cultured (cellular density $3 \times 10^5$ cells/well) and then co-infected at DIV3 with AAV9-c-Fos/ZsGreen1-DR (FLEN) and a secondary AAV9-CB7/mCherry, which constitutively expresses mCherry in infected neurons (*Figure 1B, C*). Neurons were cultured until DIV11 in regular neurobasal medium, before switching to an activating medium containing 4-aminopyridine (4-AP, 100 μM) and bicuculline (Bic, 10 μM) for 30 min. The drugs were then washed out, and tetrodotoxin (TTX, 0.5 μM) was added to block further neuronal activity until cells were fixed (*Figure 1C*). The percentage of activated neurons (ZsGreen+, further named FLEN+ neurons) over the total number of transduced neurons (mCherry+ neurons) was calculated at different time points within a 6-hr period after pharmacological stimulation. Only a limited number of FLEN+ neurons could be found in unstimulated conditions (5.4 ± 0.5 FLEN+/FLEN−, n = 3 plates) (*Figure 1D, E*). The activating medium triggered ZsGreen1 expression in neurons as early as 2 hr, reaching a threefold peak within 3 hr (1 hr: 6.7 ± 1.0, n = 3; 2 hr: 13.2 ± 5.0, n = 3; 3 hr: 15.7 ± 2.2, n = 5) (*Figure 1D, E*). The number of FLEN+ neurons decreased after the peak, decaying to pre-stimulation-like levels within 6 hr (4.5 hr: 12.0 ± 3.9, n = 4; 6 hr: 5.0 ± 1.1, n = 4) (*Figure 1D, E*). Thus, FLEN showed a rapid induction (2–3 hr) and a fast decay (within 6 hr) of ZsGreen1-DR fluorescence (one-way ANOVA, p = 0.04). To test specificity, we assessed co-expression with endogenous c-Fos in hippocampal cultures 3 hr following the same pharmacological stimulation. Among infected neurons (mCherry+), we found that 77.1 ± 6.3% (n = 4) of FLEN+ neurons were positive for c-Fos. We also found that 60.8 ± 5.4% (n = 4) of the c-Fos−+ neurons were also positive for FLEN. Thus, although FLEN does not capture the entire c-Fos+ population, FLEN+ neurons are likely true engram (c-Fos+) cells.

### FLEN captures early neuronal activity following contextual fear conditioning

We next investigated the time course of FLEN expression in vivo in wild-type mice in response to exposure to a salient novel experience. We virally targeted bilaterally dorsal CA3 in adult mice (2- to 3-month-old) co-injecting AAV9-c-Fos/ZsGreen1-DR and the infection control AAV9-CB7/mCherry (*Figure 2A*). Then, mice were subjected to contextual fear conditioning (CFC), a hippocampal-dependent one-trial memory task. During this task, mice learned to associate the delivery of three mild foot-shocks (0.4 pA, 2 s) with a specific environment (*Figure 2B*). We found that CFC, a one-trial

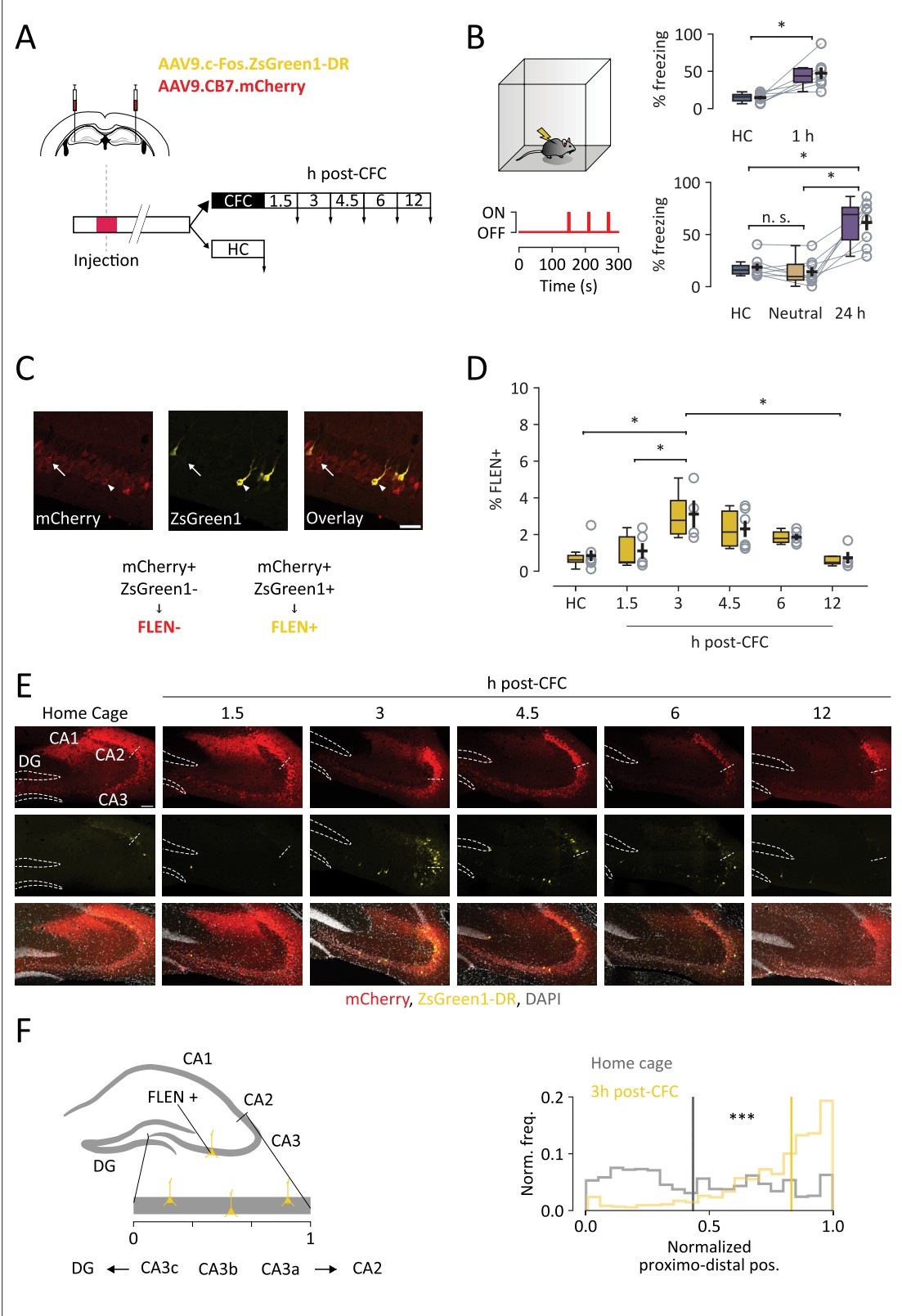

**Figure 2.** In vivo FLEN labeling and decay time course following a salient experience. (**A**) FLEN and an infection marker viral vector are injected bilaterally into CA3. Mice undergo contextual fear conditioning (CFC), and sections are collected (downward arrows) 1.5, 3, 4.5, 6, and 12 hr following CFC. As a control, sections are obtained from mice left undisturbed in their home cage (HC). (**B**) Left: one-time CFC experiment scheme. Mice explore an arena for 5 min, and three foot-shocks are given at 180, 210, and 270 s after the start of the test (red line). Top-right: paired comparison of the

*Figure 2 continued on next page*

*Figure 2 continued*

percentage of time mice spent freezing during training and when replaced in the conditioning arena 1 hr after CFC. Bottom-right: comparison of the percentage of time mice spent freezing during training, when placed in a different neutral environment and 24 hr after CFC. These experiments show that the CA3-dependent shock-context association is rapidly learned (within 1 hr), is highly environment-selective, and is retained for 24 hr. (**C**) Representative images of CA3 neurons expressing the infection marker (mCherry+, indicated by arrow) and neurons expressing both the infection marker and FLEN (ZsGreen1+/mCherry+, indicated by arrowhead). The latter are considered FLEN+ neurons. Scale bar = 50 µm. (**D**) Percentage of ZsGreen+ CA3 neurons over the total number of mCherry+ neurons at different intervals following activation. (**E**) Representative sections of CFC-trained mice compared to HC untrained mice at different intervals following CFC. Dashed lines outline the dentate gyrus (DG) cellular layer, while the dashed segment indicates the separation between CA3 and CA2. Scale bar = 100 µm. (**F**) Left panel: schematic representation of neuronal position along the proximodistal axis analysis of CA3. The CA3 pyramidal layer was straightened and the x position of FLEN+ cells was normalized on a 0 (closer to DG) to 1 (closer to CA2) scale. Right panel: comparison of normalized proximodistal frequency distribution of CA3 FLEN+ cells placement in HC mice (gray) and 3 hr post-CFC mice (yellow). Bold lines represent the median of the respective distributions. HC median is displayed in the CFC plot to highlight distribution difference. Statistical signficance: *p<0.05; ***p<0.001.

The online version of this article includes the following figure supplement(s) for figure 2:

**Figure supplement 1.** FLEN expression in different behavioral conditions.

**Figure supplement 2.** FLEN+ cells distribution along the proximodistal and superficial-to-deep axis.

experience, can generate a strong context-specific fear memory, as mice responded by freezing when re-exposed to the conditioning environment 1 hr after training (HC: 14.7% ± 2.3%, 1 hr post-CFC: 47.4% ± 7.9%, *n* = 7, unpaired *t*-test, p = 0. 0018), a response that outlasted 24 hr after training, but was not triggered in an unfamiliar novel environment, nor when they were left undisturbed in their home cage (HC) (HC: 18.7% ± 3.5%, 24 hr post-CFC: 61.5% ± 7.5%, neutral: 14.3% ± 4.5%, *n* = 8, one-way ANOVA, Tukey's post hoc test HC: 1 hr post-CFC p = 0.00004, HC: Neutral p = 0.84116, Neutral: 1 hr post-CFC p = 0.00001) (***Figure 2B***).

The percentage of FLEN+ CA3 neurons (i.e. that display FLEN+/mCherry+ labeling, ***Figure 2C***) among the overall population of infected neurons (FLEN−, i.e. that display mCherry+ labeling) was quantified 1.5, 3, 4.5, 6, and 12 hr following CFC and compared to that of untrained mice. Consistent with previous studies (***Sørensen et al., 2016***; ***Weng et al., 2018***), less than 2% of infected CA3 neurons were reported to be labeled in the absence of a novel experience (***Figure 2D, E***, HC, 0.84% ± 0.29% FLEN+/mCherry+, *n* = 6). One-trial CFC conditioning generated an engram, as indicated by an increase in FLEN+ neurons in CA3 in the following hours (***Figure 2D, E***). Specifically, CA3 displayed 1.11% ± 0.42% FLEN+ neurons 1.5 hr after CFC (*n* = 5), which increased to 3.11% ± 0.74% 3 hr post-CFC (*n* = 4) and 2.31 ± 0.44% at 4.5 hr post-CFC (*n* = 7). The fraction of active cells returned to basal-like levels at 6 hr (1.86 ± 0.16% FLEN+/ mCherry+, *n* = 5) and further decreased 12 hr post-CFC (0.73% ± 0.32% FLEN+/ mCherry+, *n* = 4) (***Figure 2D, E***). Overall, the emergence of FLEN assembly peaks 3 hr following CFC and significantly decreases 12 hr later (one-way ANOVA p = 0.003, post hoc Tukey's multiple comparisons test: HC vs. 3 hr p = 0.008, 1.5 vs. 3 hr p = 0.04, 3 vs. 12 hr p = 0.01), faithfully mimicking the progression and transient nature of native c-Fos expression (***Holtmaat and Caroni, 2016***).

We sought to probe CA3 engram formation 3 hr after different behavioral situations, namely context-only (CO) and immediate-shock (IS) conditions, which offer the possibility for spatial exploration only and no spatial exploration at all, respectively (***Figure 2—figure supplement 1A***). In CO conditions, mice were allowed to freely explore a novel cue-enriched arena. Comparably to other exploration tasks, we found an increase in the number of FLEN+ CA3 PNs 3 hr following CO (3.50 ± 1.18% FLEN+/FLEN−, *n* = 5) compared to control HC mice (Kruskal–Wallis non-parametric test, p = 0.03; Dunn post hoc multiple comparisons test, p = 0.02) (***Figure 2—figure supplement 2B***). In the IS test, a 6-s-long foot-shock was delivered 2 s after the mouse was placed in the conditioning cage, preventing sensorial sampling of the novel context (***Figure 2—figure supplement 1A***). In contrast, the IS test did not induce a significant increase in the number of FLEN+ in CA3 PNs as compared to HC (1.84 ± 0.77% FLEN+/FLEN−, *n* = 4, Dunn post hoc multiple comparisons test, HC: IS, p = 0.51) and to context-only (CO: IS, p = 0.88, *n* = 4), possibly because of minimal spatial exploration.

Because of the reported functional heterogeneity of CA3 PNs in relation to memory encoding and recall, which is directly related to their position within the proximodistal axis of CA3 (***Nakamura et al., 2013***; ***Nakazawa et al., 2016***; ***Sun et al., 2017***), we analyzed the spatial distribution of FLEN+ PNs within CA3 pyramidal layer along the proximodistal and superficial-to-deep axis of CA3, following

one-trial CFC (*Figure 2F*, *Figure 2—figure supplement 2*). We found a clear skewness in the location of FLEN+ neurons along the proximodistal axis toward CA3a at all time points following CFC and following CO compared to HC (Kolmogorov–Smirnov test: HC vs. 1.5 hr post-CFC, p < 0.001; HC vs. 3 hr post-CFC, p < 0.001; HC vs. 4.5 hr post-CFC, p < 0.001; HC vs. 6 hr post-CFC, p < 0.001; HC vs. 12 hr post-CFC, p < 0.001; HC vs. CO, p < 0.001: HC vs. IS, p < 0.001) while they appeared to be more evenly distributed along the superficial-deep axis (Kolmogorov–Smirnov test: HC vs. 1.5 hr post-CFC, p = 0.013: HC vs. 3 hr post-CFC, p < 0.001; HC vs. 4.5 hr post-CFC, p < 0.001; HC vs. 6 hr post-CFC, p < 0.001; HC vs. 12 hr post-CFC, p = 0.61; HC vs. CO, p < 0.001: HC vs. IS, p = 0.02) (*Figure 2F*, *Figure 2—figure supplement 2*). Overall, these data indicate that the FLEN labeling strategy faithfully identifies neurons activated during a one-trial exposure to a novel context and allows for the characterization of the morpho-functional properties of CA3 engram neurons shortly (<6 hr) following the experience.

In parallel, we confirmed that the RAM system (*Sørensen et al., 2016*) is a powerful tool to tag neurons selectively activated by CFC in a temporally controlled manner (*Figure 3—figure supplement 1A*; *Sørensen et al., 2016*; *Weng et al., 2018*). To identify RAM+ neurons, we stereotactically injected AAV9-RAM/tTA::TRE/EGFP and AAV9-CB7/mCherry to label dorsal CA3 in wild-type mice (*Figure 3—figure supplement 1A*). Mice were fed with doxycycline pellets starting 1 day before injection and then switched to a regular food diet for 48 hr before CFC (*Figure 3—figure supplement 1A*). As reported, we found that the number of RAM+ neurons was notably higher 24 hr after CFC with respect to HC conditions (% RAM+ vs. mCherry+ neurons, HC: 1.13 ± 0.18%, *n* = 5; CFC: 3.54 ± 0.77%, *n* = 5; Welch's *t*-test, p = 0.03) (*Figure 3—figure supplement 1B, C*), although not to the same extent as in previous reports (*Sørensen et al., 2016*). The ratio of FLEN+ neurons 3 hr after CFC over HC (approximately 3:1) closely resembles that of RAM+ neurons 24 hr after CFC (approximately 3:1).

## CA3 engram neurons evolve from low to higher firing frequency over 24 hr

To understand if rapid memory acquisition and early consolidation is supported by specific changes in electrophysiological intrinsic properties, we performed whole-cell patch-clamp recordings of CA3 PNs (mainly in CA3b) in acute hippocampal slices of young adult mice injected with either FLEN or RAM, respectively (*Figure 3A*). AAV9-c-Fos/ZsGreen1-DR or AAV9-RAM-tTA::TRE-EGFP viral constructs were injected bilaterally in dorsal CA3, together with AAV9-CB7-mCherry (*Figure 3A*). After CFC, mice were returned to their home cage for 3 hr (FLEN) or 24 hr (RAM), to yield peak expression of ZsGreen1-DR or EGFP and then sacrificed to collect parasagittal slices (*Figure 3A*). Activity-dependent FLEN+ neurons remained detectable in acute slices, enabling specific targeting of engram CA3 PNs neurons for approximately 4–5 hr after slicing. Engram neurons (FLEN+ or RAM+) were compared to their respective neighboring non-engram neurons (FLEN− or RAM−) (*Figure 3A*).

We first investigated the intrinsic physiological properties of FLEN+ and RAM+ CA3 PNs. Indeed, the probability for a neuron to be recruited in a memory engram is thought to depend on its inherent excitability level at the time of learning (see *Josselyn and Tonegawa, 2020* for review). We assessed input resistance ($R_i$) by injecting a series of hyper- and depolarizing current pulses with the membrane potential held at –70 mV (*Figure 3B*). $R_i$ did not change significantly between FLEN+ and FLEN− neurons (FLEN+: 253 ± 35 MΩ, *n* = 19; FLEN−: 210 ± 19 MΩ, *n* = 30; non-parametric Mann–Whitney *U* test, p-value = 0.57) and was similar to RAM+ and RAM− neurons which were also no different between them (RAM+: 323.7 ± 14.1 MΩ, *n* = 21; RAM−: 396.1 ± 15.2 MΩ, *n*=30, parametric *t*-test, p = 0.68) (*Figure 3B*). Further depolarizing steps of current were injected until an action potential (AP) was fired. AP properties such as threshold, amplitude, width, upstroke, and downstroke were not statistically different between FLEN+ and FLEN− neurons and between RAM+ and RAM− neurons (*Figure 3—figure supplement 2*; *Table 1*). To study the AP firing pattern, we injected 1-s-long pulses of current just above rheobase (200 pA) (*Figure 3C*). Overall, the firing frequency was not different in the FLEN groups (FLEN+: 2.64 ± 0.65 Hz, FLEN−: 3.59 ± 0.62 Hz, Mann–Whitney *U* test, p = 0.51) (*Figure 3C*). More than half of the total number of APs occurred within the first 100 ms for all CA3 PNs, showing a consistent discharge pattern across both FLEN+ and FLEN− cells (Kolmogorov–Smirnov test, p-value = 1.0) (*Figure 3D*). In contrast to FLEN+ CA3 PNs, RAM+ CA3 PNs showed a prolonged spiking activity which outlasted the first 100 ms (*Figure 3D*). Altogether, these results show that early after encoding, FLEN+ CA3 PNs do not show differences in excitability or patterns of

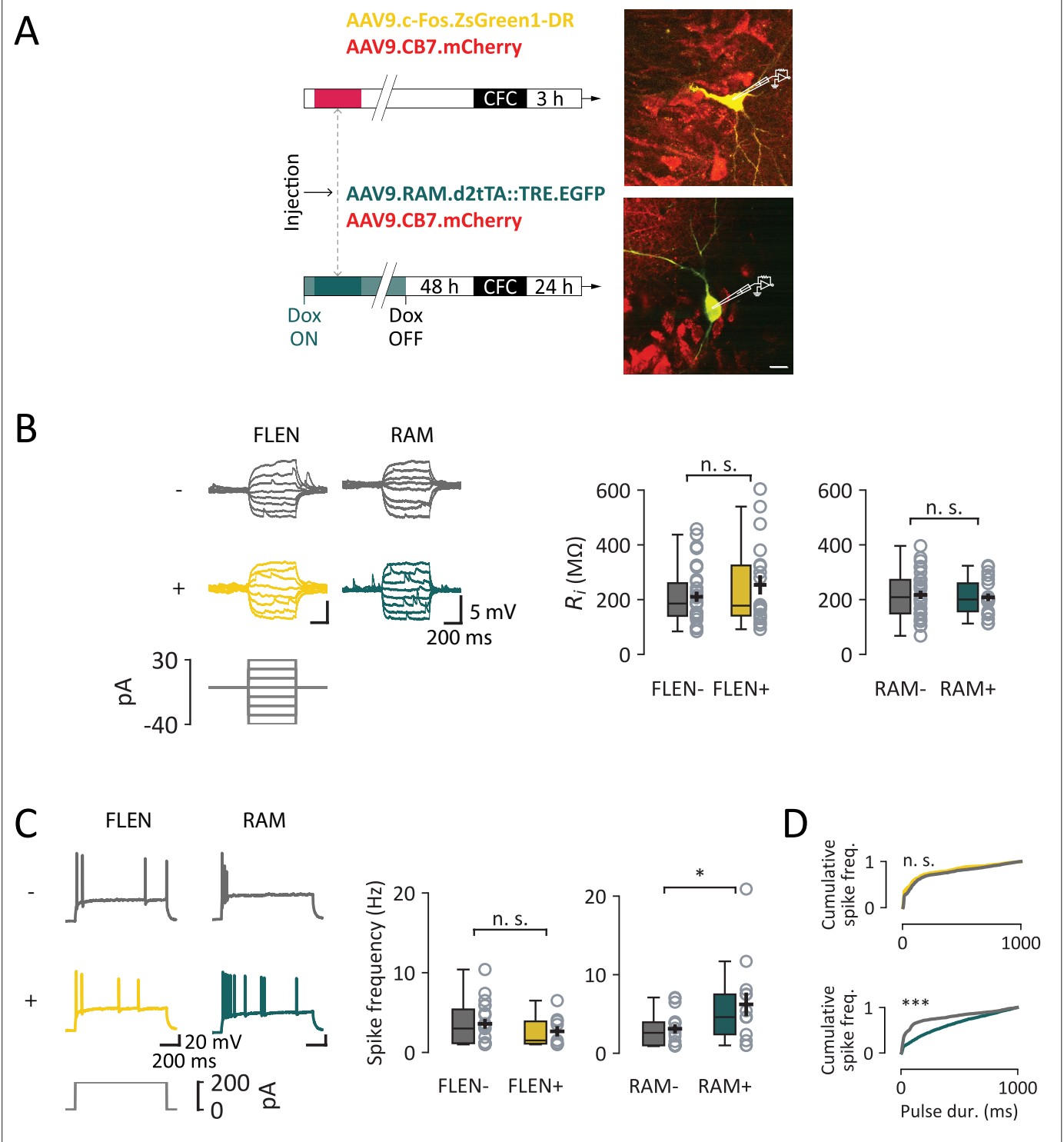

**Figure 3.** Electrophysiological analysis of intrinsic properties of CA3 engram neurons. (**A**) Experimental outline: mice were injected with either FLEN or RAM construct, subjected to contextual fear conditioning (CFC) and then sacrificed at either 3- or 24-hr post-conditioning to collect acute hippocampal slices. Right: representative images of FLEN+ and RAM+ cells targeted for patch-clamp recordings, with neighboring control FLEN- or RAM− neurons. Scale bar = 20 µm. (**B**) Membrane potential response. Left panel: representative membrane potential trace in response to incremental injected current in control cells (gray) and engram cells (yellow for FLEN and green for RAM). Right panel: graphs of input resistance for comparison between engram cells and their corresponding control cells. (**C**) Action potential firing pattern. Left panel: sample traces of long (1-s) above-rheobase current step to analyze the pattern of action potential firing current in control cells (gray) and engram cells (yellow and green). Right panel: average spike frequency graphs.

*Figure 3 continued on next page*

*Figure 3 continued*

(**D**) Cumulative spike frequency curve over the 1-s current step shown in (**C**) showing a similar distribution between FLEN + and FLEN− neurons (top), and a more sustained firing of RAM+ compared to their RAM− neurons (bottom). Statistical signficance: *p<0.05; ***p<0.001.

The online version of this article includes the following figure supplement(s) for figure 3:

**Figure supplement 1.** RAM expression 24 hours following CFC.

**Figure supplement 2.** Action potential properties of FLEN+, FLEN-, RAM+ and RAM-neurons.

activity as compared to control neurons. However, after 24 hr, CA3 engram cells display an increase in excitability without changes in $R_i$.

## Synaptic properties of FLEN+ neurons

To assess overall excitatory synaptic strength and overall number of active synaptic connections, we recorded spontaneous excitatory post-synaptic currents (sEPSCs) in FLEN+ CA3 PNs while blocking $GABA_A$-dependent inhibitory currents (bicuculline, 10 μM). The average amplitude of sEPSCs was not different between FLEN+ and FLEN− CA3 PNs (FLEN+: 29.7 ± 5.3 pA, *n* = 10, FLEN−: 32.2 ± 3.7 pA, *n* = 10, Kolmogorov–Smirnov test, p=0.99), whereas their frequency was increased (inter-event interval (IEI), FLEN+: 788 ± 152 ms, *n* = 10, FLEN−: 930 ± 91 ms, *n* = 10, Kolmogorov–Smirnov test, p < 0.005; frequency: FLEN+: 1.64 ± 0.26 Hz, *n* = 10, FLEN−: 1.19 ± 0.14 Hz, *n* = 10, pair-wise Mann–Whitney *U* test, p = 0.27) (*Figure 4A*). Similarly, 24 hr after a novel experience, the average amplitude of sEPSCs was similar between RAM+ and RAM− CA3 PNs (RAM+: 26.22 ± 2.46 pA, *n* = 14; RAM−: 21.31 ± 2.72 pA, *n* = 14, Kolmogorov–Smirnov test, p = 0.99) (*Figure 4A*), but their frequency was significantly increased in RAM+ CA3 PNs (IEI: RAM+: 645 ± 88 ms, *n* = 17; RAM−: 1342 ± 207 ms, *n* = 17; Kolmogorov–Smirnov test, p < 0.05; frequency: RAM+: 2.04 ± 0.26 Hz, *n* = 17, RAM−: 0.93 ± 0.11 Hz, *n* = 14, pair-wise Mann–Whitney *U* test, p = 0.001) (*Figure 4A*).

Then, to investigate excitatory synaptic strength and release probability independent of network activity, we measured miniature EPSCs (mEPSCs) by further adding TTX (0.5 μM). The average amplitude was similar between FLEN+ and FLEN− neurons (FLEN+: 20.5 ± 2.5 pA, *n* = 9, FLEN−: 18.7 ± 1.6 pA, *n* = 14, Kolmogorov–Smirnov test, p = 0.99), whereas we observed a higher frequency of mEPSCs in FLEN+ neurons as compared to control neurons (IEI: FLEN+: 1228 ± 186 ms, *n* = 9, FLEN−: 1722 ± 242 ms, *n* = 14, Kolmogorov–Smirnov test, p < 0.0005; frequency: FLEN+: 1.48 ± 0.66 Hz, *n* = 9, FLEN−: 0.69 ± 0.07 Hz, *n* = 14, pair-wise Mann–Whitney *U* test, p = 0.29) (*Figure 4B*). Comparably, we found a higher frequency of mEPSCs in RAM+ neurons as compared to RAM− CA3 PNs (IEI: RAM+: 782 ± 91 ms, *n* = 14; RAM−: 1244±247 ms, *n* = 13; Kolmogorov–Smirnov test, p < 0.005;

**Table 1.** Summary of AP properties.

| | FLEN | | | RAM | | |
|---|---|---|---|---|---|---|
| | FLEN− | FLEN+ | Test | RAM− | RAM+ | Test |
| Peak (mV) | 45.19 ± 2.25, *n* = 19 | 43.92 ± 4.62, *n* = 9 | Mann–Whitney *U*, p = 1.00 | 47.93 ± 1.41, *n* = 15 | 50.91 ± 1.57, *n* = 13 | *T*-test, p = 0.17 |
| Threshold (mV) | −40.41 ± 1.54, *n* = 20 | −42.47 ± 1.36, *n* = 14 | *T*-test, p = 0.35 | −41.53 ± 1.35, *n* = 15 | −39.78 ± 2.46, *n* = 13 | *T*-test, p = 0.52 |
| Amplitude (mV) | 99.91 ± 4.63, *n* = 19 | 98.67 ± 8.73, *n* = 9 | *T*-test, p = 0.89 | 103.71 ± 3.93, *n* = 15 | 101.18 ± 4.53, *n* = 13 | *T*-test, p = 0.67 |
| Width (ms) | 0.002 ± 0.001, *n* = 16 | 0.002 ± 0.001, *n* = 6 | Mann–Whitney *U*, p = 0.51 | 0.0017 ± 0.0001, *n* = 13 | 0.0016 ± 0.00003, *n* = 7 | Mann–Whitney *U*, p = 0.36 |
| Upstroke (mV/s) | 197.06 ± 19.54, *n* = 19 | 213.35 ± 35.48, *n* = 9 | *T*-test, p = 0.67 | 229.89 ± 17.86, *n* = 15 | 241.65 ± 23.2, *n* = 13 | *T*-test, p = 0.69 |
| Downstroke (mV/s) | −57.99 ± 7.91, *n* = 19 | −54.99 ± 6.18, *n* = 9 | Mann–Whitney *U*, p = 0.84 | −56.70 ± 4.43, *n* = 15 | −59.96 ± 3.62, *n* = 13 | *T*-test, p = 0.58 |
| Initial frequency (Hz) | 21.58 ± 4.15, *n* = 16 | 25.04 ± 12.83, *n* = 7 | Mann–Whitney *U*, p = 0.62 | 41.74 ± 9.54, *n* = 10 | 25.30 ± 6.43, *n* = 11 | Mann–Whitney *U*, p = 0.25 |
| ISI (ms) | 0.19 ± 0.05, *n* = 16 | 0.26 ± 0.08, *n* = 7 | Mann–Whitney *U*, p = 0.28 | 0.14 ± 0.02, *n* = 10 | 0.14 ± 0.02, *n* = 11 | *T*-test, p = 0.98 |

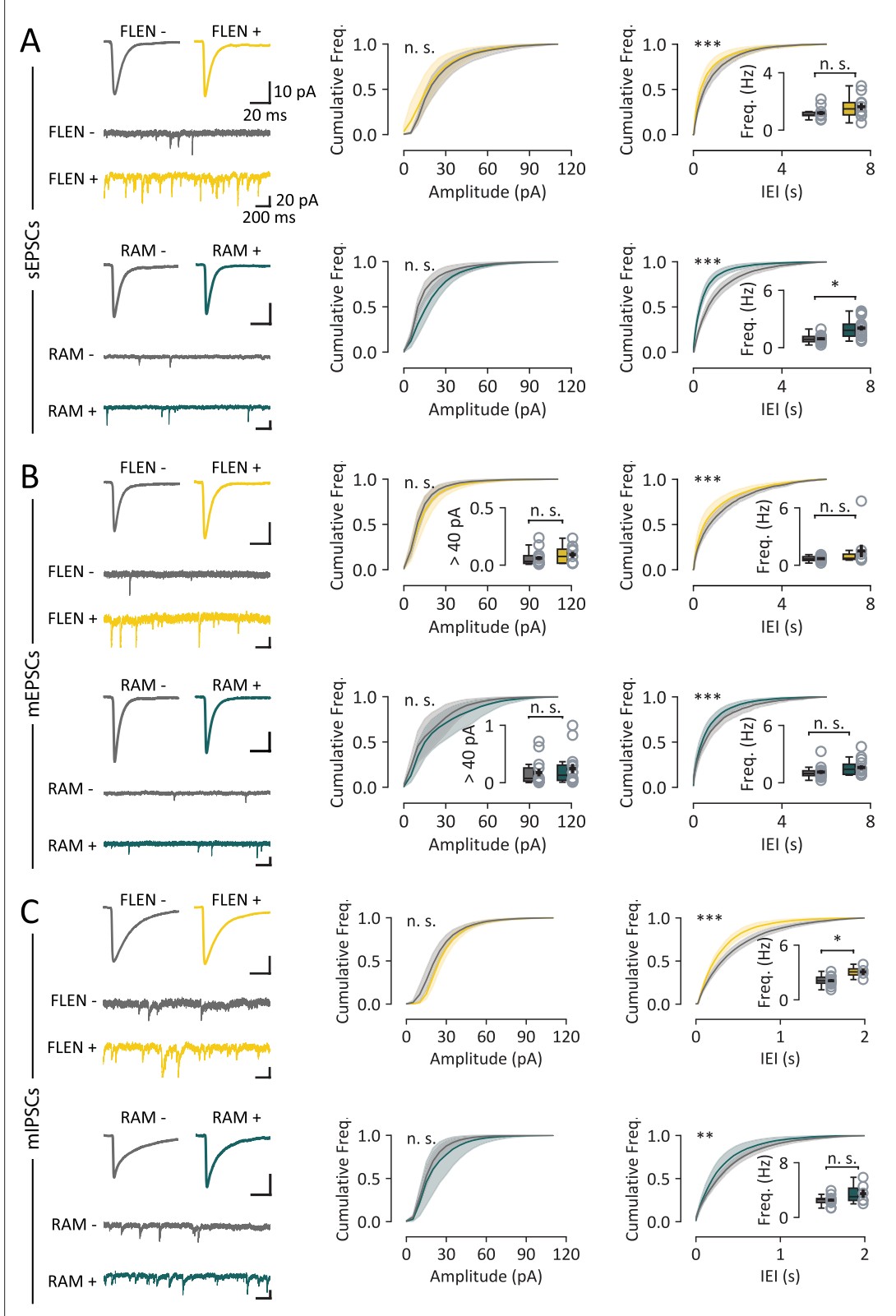

**Figure 4.** Analysis of EPSCs and IPSCs in CA3 engram neurons. (**A**) Sample traces and cumulative distributions of spontaneous excitatory post-synaptic currents (sEPSCs) recorded from CA3 engram neurons (FLEN in yellow, RAM in green). Top panels: comparison between FLEN+ and FLEN− neurons reports no difference in mean sEPSC amplitude cumulative distribution but a higher event frequency in FLEN+ neurons, as seen in the inter-event interval (IEI) cumulative distribution. The inset displays the comparison of the average frequency (Hz) of such events. Bottom panels: RAM+ vs. RAM−

*Figure 4 continued on next page*

*Figure 4 continued*

neurons show a similar pattern characterized by no change in amplitude and increased frequency in RAM+ neurons. (**B**) Same analysis as in (**A**), for miniature EPSCs (mEPSCs). The inset in the amplitude plot shows the fraction of mEPSCs exceeding 40 pA, indicative of giant mossy fiber events. (**C**) As (**A**) and (**B**), for miniature IPSCs (mIPSCs), comparing frequency and amplitude in FLEN+ vs. FLEN− and RAM+ vs, RAM− neurons. Statistical signficance: *p<0.05; ** p<0.01; ***p<0.001.

frequency: FLEN+: 1.60 ± 0.23 Hz, *n* = 14, FLEN−: 1.12 ± 0.20 Hz, *n* = 13, pair-wise Mann–Whitney *U* test, p = 0.08), with unchanged amplitude (RAM+: 29.6 ± 4.8 pA, *n* = 14, RAM−: 24.7 ± 4.2 pA, *n* = 13, Kolmogorov–Smirnov test, p = 0.99) (*Figure 4B*). Thus, neurons recruited in an engram appear to show increased excitatory drive after a few hours that persist after 24 hr, which may be linked to a higher number of synaptic contacts.

Previous reports indicate that CA3 PNs activated during contextual learning and identified 24 hr later by RAM had strengthened Mf inputs (*Weng et al., 2018*). The existence of mEPSCs with larger amplitudes may be related to synaptic plasticity in a subset of synapses, or theoretically to an increased participation of Mf-driven mEPSCs (*Henze et al., 2002*). We thus calculated the proportion of miniature excitatory events with an amplitude greater than 40 pA. During spontaneous activity recording, we found no difference in the fraction of putative Mf-driven inputs between FLEN+ and FLEN− PNs (FLEN+: 0.09 ± 0.03%, *n* = 9; FLEN−: 0.06 ± 0.02%, *n* = 14; Mann–Whitney *U* test, p = 0.33) and between RAM+ and RAM− PNs 24 hr after learning (RAM+: 0.25 ± 0.08%, *n* = 14; RAM−: 0.18 ± 0.07%, *n* = 13; Mann–Whitney *U* test, p = 0.42) (*Figure 4B*, inset). This later finding contrasts with earlier findings (*Weng et al., 2018*), which found a selective strengthening of Mf inputs onto the ensemble of CA3 neurons >24 hr after activation by CFC. At variance with our analysis, this later study uses a mGluR2/3 agonist, LY354740, which inhibits evoked Mf-CA3 synaptic transmission (*Kamiya and Ozawa, 1999*).

Inhibitory neurons are important in controlling spike generation of CA3 PNs, shaping network activity and participation in neuronal assembly formation (*Holtmaat and Caroni, 2016*). We therefore probed miniature inhibitory post-synaptic currents (mIPSCs), comparing FLEN+ vs. FLEN− neurons and RAM+ vs. RAM− neurons, in the presence of NBQX (20 µM), D-AP5 (50 µM), and TTX (0.5 µM). We found that mIPSCs were statistically more frequent (IEI: FLEN+: 340.1 ± 4.0 ms, *n* = 4, FLEN−: 511.0 ± 40.5 ms, *n* = 14, Kolmogorov–Smirnov test, p < 0.005; frequency: FLEN+: 3.06 ± 0.35 Hz, *n* = 4, FLEN−: 2.09 ± 0.14 Hz, *n* = 14, pair-wise Student's *t*-test, p = 0.008), but their amplitude was on average comparable in FLEN+ vs. FLEN− neurons (FLEN+: 35.1 ± 2.3 pA, *n* = 4, FLEN+: 31.7 ± 2.2 pA, *n* = 14, Kolmogorov–Smirnov test, p = 0.99, Welch's *t*-test, p = 0.30) (*Figure 4C*). At 24 hr after learning, RAM+ neurons showed an increase in the frequency of mIPSC (IEI: 330.5 ± 51.9 ms, *n* = 6; frequency: 3.48 ± 0.60 Hz, *n* = 6) compared to RAM− neurons (IEI: 437.1 ± 47.1 ms, *n* = 11, Kolmogorov–Smirnov test, p < 0.005; frequency: 2.52 ± 0.23 Hz, *n* = 11, pair-wise Mann–Whitney *U* test, p = 0.09), but no difference in their amplitude (RAM+: 22.7 ± 1. 8 pA, *n* = 11; RAM−: 26.5 ± 5.2, *n* = 6, Kolmogorov–Smirnov test, p = 0.89) (*Figure 4C*). The frequency of all excitatory and inhibitory inputs was elevated after CFC, both after 3 hr and after 24 hr, but not their amplitude.

## Mf-CA3 synapses and feedforward inhibition in FLEN+ neurons

Mf-CA3 connections between the DG and CA3 pyramidal cells provide powerful inputs via 'giant' Mf boutons to CA3 PCs, which are thought to assist CA3 in the encoding of memory (*Marneffe et al., 2025*; *Vandael and Jonas, 2024*). We recorded EPSCs evoked by minimal stimulation of Mfs (*Marchal and Mulle, 2004*) in FLEN+ and FLEN− CA3 PNs using an extracellular electrode placed within the DG hilus to stimulate the Mf axonal bundle (*Figure 5A*). Low-frequency stimulation (0.1 Hz) evoked EPSCs of variable amplitude that, on average, did not differ between FLEN+ (81.7 ± 17.5 pA) and FLEN− neurons (80.8 ± 13.5 pA, Welch's *t*-test, p = 0.97) (*Figure 5B*). Increasing stimulation frequency to 1 Hz induced short-term presynaptic frequency facilitation, causing an approximately threefold increase in evoked current amplitude in both FLEN+ (258.3 ± 49.5 pA) and FLEN− neurons (238.2 ± 39.0 pA, Welch's *t*-test, p = 0.76) (*Figure 5B*). The EPSC facilitation ratio (1 vs. 0.1 Hz) was not different between FLEN+ and FLEN− neurons (FLEN+: 3.28 ± 0.40, FLEN−: 3.28 ± 0.57, Welch's *t*-test, p = 0.99) (*Figure 5B*).

One-trial memory tasks reversibly induce a robust and rapid (within hours) increase in the number of filopodia emanating from Mf synaptic terminals which contact GABAergic interneurons (*Ruediger*

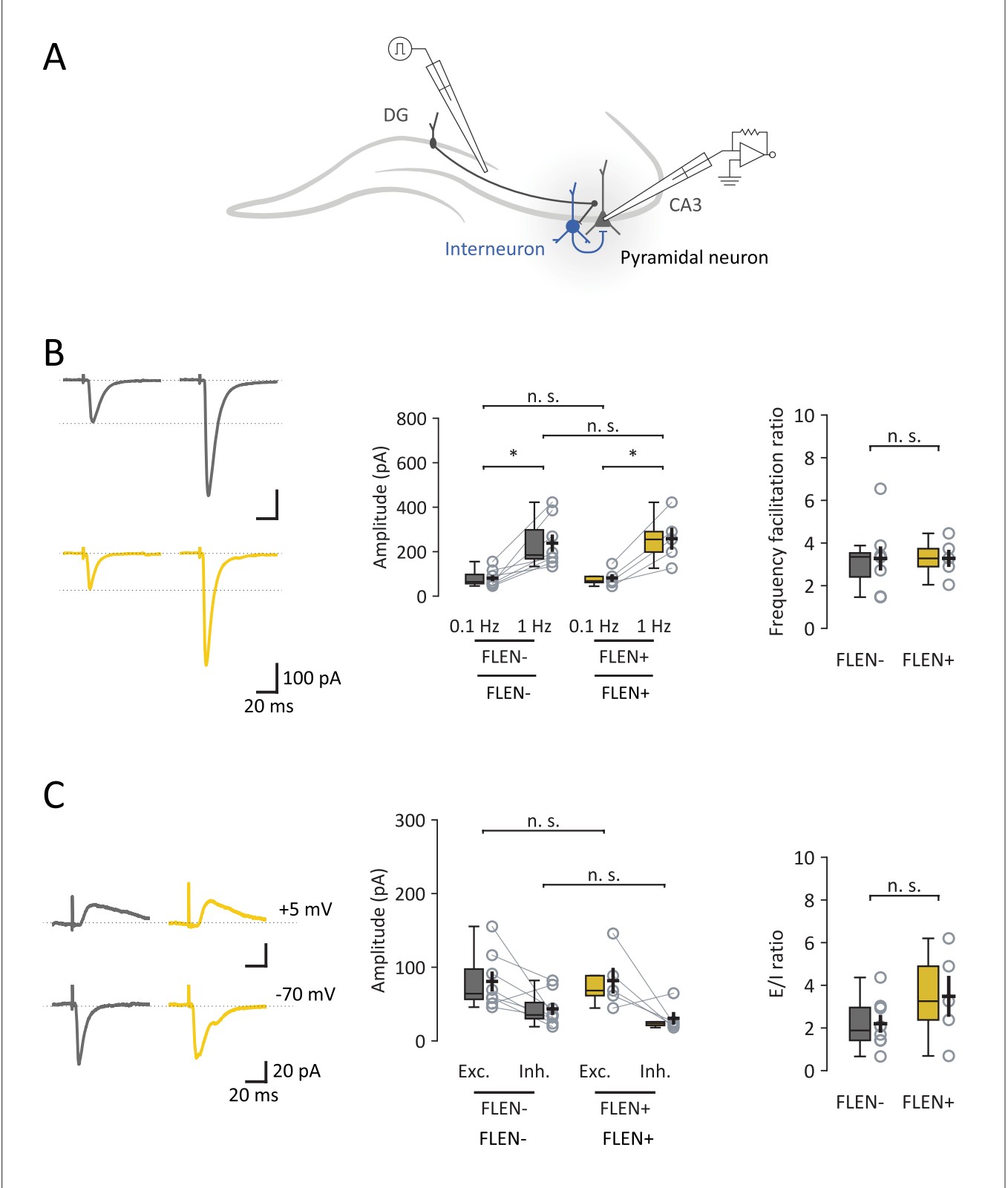

**Figure 5.** 5Properties of Mf-CA3 synapses in CA3 engram neurons. (**A**) Schematic representation of Mf-CA3 EPSCs and feedforward inhibition recordings. (**B**) Left panel: representative traces showing EPSC amplitude in FLEN− (gray) and FLEN+ (yellow) cells following low frequency stimulation (0.1 Hz) and corresponding increase evoked with a higher stimulation frequency (1 Hz), due to a presynaptic form of synaptic plasticity characteristics of Mf-CA3 synapses (frequency facilitation). Center panel: amplitude plot for FLEN− and FLEN+ stimulated at 0.1 and 1 Hz. Right panel: frequency

*Figure 5 continued on next page*

*Figure 5 continued*

facilitation ratio plot shows no difference between FLEN− and FLEN+ neurons. (**C**) Left panel: representative traces showing Mf-driven IPSCs (feedforward inhibition) and EPSCs (direct excitatory activity) recorded in individual FLEN− (gray) and FLEN+ (yellow) neurons. Center panel: amplitude plot of EPSCs and IPSCs for FLEN− and FLEN+ neurons. Right panel: graph showing excitation-to-inhibition ratio, comparing FLEN− and FLEN+ neurons. Statistical signficance: *$p < 0.05$.

*et al., 2011*), lending support to the hypothesis of an increase in feedforward inhibition in most CA3 PNs. Because of the rapid time course with which structural plasticity occurs at Mf-CA3 synapses following one-trial memory task, we took advantage of FLEN to explore a functional counterpart to structural plasticity, and to address whether the change in feedforward inhibition was differentially affected in FLEN+ vs. FLEN− CA3 PNs. We recorded excitatory and inhibitory synaptic currents in CA3 PNs while stimulating Mfs (*Figure 5A*). To calculate the excitatory to inhibitory (E/I) ratio at these synapses, we first recorded evoked EPSCs by holding cells at the experimentally determined GABA$_A$ reversal potential ($E_{Cl−} = −70$ mV), and then at the reversal potential for cations ($E_{Na1+} = +5$ mV) (*Figure 5C*; *Torborg et al., 2010*). We measured the peak amplitude of evoked EPSCs and IPSCs at a stimulation rate of 0.1 Hz to prevent frequency facilitation (*Figure 5C*). We found that the EPSC/IPSC ratio was higher in FLEN+ vs. FLEN− cells, although this is not significantly different (FLEN+: 3.48 ± 0.96, FLEN−: 2.20 ± 0.42, Welch's *t*-test, $p = 0.27$) (*Figure 5C*). These data do not support the notion that shortly after one-trial learning, there is an increase in feedforward inhibition at Mf-CA3 synapses.

## Discussion

In the past decade, the search for engrams and the characterization of engram cells has largely benefited from new gene targeting approaches to label the cells and to control their reactivation (*Josselyn and Tonegawa, 2020*). The primary goal of our work was to test the hypothesis that the functional properties of hippocampal cells activated during a one-trial contextual memory task were distinct from surrounding ones shortly (3–6 hr) after encoding and were subject to progressive plastic changes which could be linked to a first phase of memory consolidation.

Most techniques currently employed to study the properties of engram cells ex vivo combine promoter expression based on IEGs, intersectional transgenic strategies, optogenetics, and pharmacogenetics with behavioral paradigms (*Denny et al., 2014*; *Pignatelli et al., 2019*; *Sun et al., 2020*; *Tonegawa et al., 2015*). A major caveat for using activity-dependent promoter/enhancers is the background basal expression which can result in the labeling of neurons unrelated to the triggering behavioral event. To limit this, fluorescent marker and/or actuator expression can be temporally controlled by directly coupling IEG promoters, such as c-Fos or synthetic promoters (RAM), to inducible systems such as the doxycycline (Dox) or the tamoxifen (TAM) systems. With these approaches, the physiological properties of engram vs. non-engram cells have been examined by ex vivo patch-clamp recordings 24 hr to several days after the initial contextual memory task (*Pignatelli et al., 2019*; *Sørensen et al., 2016*; *Weng et al., 2018*; *Yap et al., 2021*).

We thus developed a viral strategy for transient neuronal expression of a fluorescent protein (FLEN) following the exposure of mice to a novel context making use of the full-length c-Fos promoter and of the brightly fluorescent ZsGreen1 protein (8.6 times brighter than EGPF, *Bell et al., 2007*), engineered to be quickly destabilized. The time course of expression of ZsGreen1 in FLEN-labeled neurons peaked at 3 hr both after pharmacological activation in cultured neurons and in vivo after a 5-min exposure to a novel context. Neuron labeling persisted for a few hours, enabling ex vivo electrophysiological characterization of neurons thought to be involved in the initial encoding of contextual memory. As with all techniques based on IEG promoters, a small percentage of AAV-transduced neurons showed ZsGreen1 fluorescence under control conditions (HC or no activation of cultured neurons). However, following CFC, the number of FLEN-labeled neurons increased by a factor of 3.12. The bright fluorescence of ZsGreen1 enabled the electrophysiological characterization of FLEN+ neurons, the majority of which (>60%) were activated in relation to the CFC memory task. Therefore, the key features of the FLEN strategy as compared to other tools used to label engrams are that: (1) it uses an AAV-based viral tool, (2) a c-fos promoter drives the expression of a brightly fluorescent protein allowing for their identification ex vivo for functional analysis, and (3) the fluorescent protein is rapidly destabilized, providing the possibility to label neurons only a few hours after their activation by a behavioral task.

We have used FLEN to label neurons in CA3, which is involved in the rapid encoding of new spatial and contextual information (*Kesner and Rolls, 2015*). One of the key features of CA3 is the presence of recurrent excitatory connections which are subject to NMDA-dependent synaptic plasticity (*Rebola et al., 2017*), and this appears essential for memory of a one-trial experience (*Nakazawa et al., 2003*). It has been proposed that the excitability state of neurons may contribute to memory formation (*Zhang and Linden, 2003*). We first examined whether FLEN+ neurons showed distinctive intrinsic excitability properties. We observed no difference in the membrane potential, input resistance, threshold, or bursting activity between FLEN+ and FLEN− neurons in slices prepared 3 hr following CFC. This suggests that the subpopulation of CA3 PNs engaged in a one-trial learning task is not initially more excitable than the general population of neurons, hence is not predetermined by their excitability state. Alternatively, engram neurons may be transiently more excitable, at the time of encoding, and lose this property within a few hours. In apparent contrast with our finding that FLEN+ CA3 PNs did not show increased excitability, a simple associative memory task, trace eye-blink conditioning, appears to increase the excitability of a large fraction (more than half) of CA3 neurons starting at 1 hr and decreasing after several days (*Thompson et al., 1996*). Beyond the difference in the task and in the species, these contrasting results may be explained by an overall increased neuronal activity related to the learning task (*Cai et al., 2016*; *Thompson et al., 1996*), however not specific to engram neurons. This does not preclude, however, the possibility that relative neuronal excitability immediately before training could contribute to the selection of neurons to an engram as proposed in the amygdala (*Yiu et al., 2014*) and the hippocampus (*Mocle et al., 2024*). However, in our hands, a one-trial memory task does not per se trigger a selective increase in the excitability of engram neurons 3 hr after CFC as compared to the general population of CA3 neurons. In the DG, recall of a previously stored contextual memory (48 hr before), hence reactivation of engram neurons, rapidly increased the excitability of these engram neurons (within 1 hr) with respect to engram neurons which were not subject to reactivation in a separate group of mice (*Pignatelli et al., 2019*). A learning-related task may, however, induce plasticity of intrinsic excitability of engram neurons on a longer time scale. We thus compared the intrinsic excitability of FLEN+ neurons to those labeled more than 24 hr after CFC by using the previously described RAM (*Sørensen et al., 2016*). We found that RAM+ neurons showed prolonged spike firing in response to a depolarizing pulse of current, hence decreased accommodation in comparison with non-labeled neurons. Considering that the set of neurons labeled by these two strategies may be of the same set, based on the expression of a fluorescent marker under the control of the c-Fos promoter, our data strongly suggest that engram neurons in CA3 progressively increase excitability as compared to neurons which were not activated by the one-trial contextual memory task. As a limitation point, it should however be considered that both strategies may label distinct populations of neurons, due to the use of different promoters to drive the expression of the fluorescent marker. However, both strategies accurately label neurons that are, in large part, c-Fos positive and are involved in processing specific context information (see *Figure 4*, *Sørensen et al., 2016*). The acquisition of increased excitability 24 hr after one-trial memory encoding suggests that intrinsic excitability is a key feature of an early consolidation of memory, which may even be strengthened upon reactivation of memory (*Pignatelli et al., 2019*). It would be interesting to assess whether increased intrinsic excitability persists throughout further consolidation of memory.

Increased neuronal activity in engram neurons at the time of encoding may result from more efficient or more synchronous excitatory synaptic inputs to these neurons. Here we observed that FLEN+ neurons (hence characterized >3 hr after initial encoding) showed an increased excitatory drive, which may be explained by a higher number of synaptic contacts, although an increase in release probability cannot be excluded. We cannot discriminate whether the increased number of synaptic inputs preexists to the formation of the engram, or whether it is a plastic mechanism taking place shortly following the contextual memory. The time course of appearance of additional synapses, by unmuting of silent synapses following contextual memory acquisition, is compatible with experimentally induced functional and structural synaptic plasticity (*Holtmaat and Caroni, 2016*). On the other hand, if FLEN+ neurons showed higher synaptic inputs at the time of memory encoding, this would increase the chance that these neurons show increased excitation (c-Fos activation) and hence participate in the engram. The amplitude of mEPSCs in FLEN+ neurons does not appear to increase on average in comparison with FLEN− CA3 PNs. This finding may not favor the possibility that LTP-like synaptic plasticity occurs early during the process of memory formation. In this event, NMDA-dependent forms

of synaptic plasticity are more likely to occur later on during the early phase of consolidation of the memory trace (*Humeau and Choquet, 2019*). Interestingly, we observed a comparable increase in the frequency of sEPSCs and mEPSCs in RAM+ CA3 neurons 24 hr after CFC, as previously reported (*Weng et al., 2018*), without any change in the amplitude. Using a pharmacological agent which selectively inhibits Mf-CA3 transmission, it was hypothesized that the increased mEPSC frequency is mainly due to more active Mf-CA3 synapses (*Weng et al., 2018*). However, if this was the case, we would also expect an increase in the amplitude of mEPSCs as well as an increase in the fraction of mEPSCs larger than 45 pA. In addition, we did not find evidence for an increase in Mf-CA3-evoked EPSCs (see below). Overall, our data indicate that CA3 engram neurons characterized a few hours after the one-trial memory task show a higher level of excitatory synaptic inputs, and this feature seems to extend to 24 hr. We found no evidence for increased amplitude of EPSCs in FLEN+ or RAM+ CA3 PNs which could have been indicative of postsynaptic plasticity through an increased number of synaptic AMPA receptors (*Humeau and Choquet, 2019*; *Takeuchi et al., 2014*). One reason may be that synaptic plasticity only occurs in a subset of synaptic contacts, and the recording of mEPSCs is not sensitive enough to capture these modifications.

Local inhibition controls the spiking of CA3 PNs, hence their capacity to be engaged in an active ensemble of neurons at the time of contextual memory encoding. In CA1, there is a fine control of inhibition upon novel environment exploration (*Yap et al., 2021*) or fear learning (*Lovett-Barron et al., 2014*). We found an increase in the frequency of mIPSCs after CFC in engram CA3 neurons, both after 3 hr and after 24 hr. This suggests that the elevation of mEPSCs in the engram neurons is counterbalanced by increased inhibition, although this would need to be generally tested in conditions of hippocampal circuit activity in vivo. We have also provided information on the process of Mf-driven feedforward inhibition in CA3 following one-trial memory encoding. Mf-CA3 synapses have been proposed to play a major role in assisting CA3 in the encoding of memory (*Kesner and Rolls, 2015*). For these reasons, it is important to directly characterize the properties of Mf-CA3 synapses impinging on CA3 engram neurons. We observed no difference in the amplitude and short-term plasticity of Mf-CA3 EPSCs evoked in FLEN+ CA3 neurons vs. FLEN- neurons. It has been proposed that 24 hr after CFC, the increased amplitude of mEPSCs in RAM+ neurons can be attributed to modifications of Mf-CA3 synapses (*Weng et al., 2018*), which may suggest progressive plasticity of these inputs. There is abundant evidence that the structural properties of Mf-CA3 synapses are modified by experience (*Ruediger et al., 2011*; *Galimberti et al., 2006*; *Maruo et al., 2016*). Feedforward inhibition combines with short-term plasticity of Mf-CA3 synapses to define the physiological conditions for efficient spike transfer between the DG and CA3 (*Marneffe et al., 2025*; *Vandael and Jonas, 2024*; *Zucca et al., 2017*). CFC induces a robust and global increase in the number of filopodia emanating from the giant Mf terminals, which peaks between 1 and 5 hr after the one-trial memory task and is related to the precision of memory and the size of engrams (*Ruediger et al., 2011*). Because filopodia contact GABAergic interneurons, this structural plasticity has been proposed to increase feedforward inhibition, albeit this has not yet been demonstrated. Here we compared the excitation/inhibition ratio of Mf-evoked synaptic responses between FLEN+ and FLEN− neurons. We did not observe a significant increase of feedforward inhibition onto CA3 engrams, as may be expected from an increased number of filopodia, but we rather observed a tendency toward less feedforward inhibition. This leads us to speculate that shortly following contextual memory formation, Mf-driven feedforward inhibition is increased in the general population of neurons but not in Mf inputs onto engram neurons, thus allowing for a comparatively stronger input from the DG to the engram neurons. It will be interesting to test whether this increased signal to noise ratio is maintained 24 hr after contextual memory encoding, given that the number of filopodia per Mf bouton has returned to basal level (*Ruediger et al., 2011*).

In conclusion, we used a viral tool allowing for the identification of neurons involved in a behavioral task within 3 hr, as a complement to the transgenic mouse strategies and viral vectors which have been used to assess changes in engram neuron properties in the time range of days. Although it is not possible to fully assess the electrophysiological properties of engram neurons at the time of the encoding, this work shows that the engram neurons are not different in their intrinsic membrane properties shortly after encoding. This set of properties may, however, follow delayed (progressive) plastic changes which are evident 24 hr after initial encoding. Overall, we propose that engram neurons do not show higher excitability preexisting to the selection of neurons to the engram during the memory

encoding task. Hence, the increased activation state of the FLEN+ neurons at the time of encoding may rather relate to a higher number of active synaptic inputs providing more efficient excitation. Interestingly, although more work is needed to confirm this tendency, increased excitation may also be linked to decreased Mf-driven feedforward inhibition, thus favoring spike transfer from the DG to CA3 at the time or shortly after memory encoding. The FLEN viral tool will be useful to assess how the properties of neurons in brain regions are involved in the contextual memory encoding process, including the entorhinal cortex and the DG.

## Materials and methods

### Animals

C57BL/6J female mice were obtained from Janvier and cared for according to the regulations of the University of Bordeaux/CNRS Animal Care and Use Committee (agreement number, APAF-IS#8444-2016051911442448). Animals were housed with their littermates with ad libitum access to food and water. Cages were kept in a temperature-regulated room on a 12-hr light/dark cycle. Prior to any behavioral test, all mice were single housed for 4–5 days and then handled for at least 3 consecutive days. Mice injected with the RAM system were kept on a doxycycline diet (4 mg/kg) starting 1 day before the viral injection.

### Viral vectors

Adeno-associated virus (AAV) serotype 2/9 vectors containing our custom-made c-Fos-ZsGreen1-DR (titer $4.00 \times 10^{13}$) were generated in our lab. The c-Fos promoter was subcloned from the Addgene_11983 construct (*Mus musculus*, size: 466 pb, position –380/+86). The AAV2/9 vectors containing the RAM system (titer $3.39 \times 10^{13}$) were purchased from AddGene (ID 84469). Each vector was mixed with an infection marker virus AAV2/9-CB7/mCherry (titer $1.50 \times 10^{13}$) and diluted 1:10 for co-injection.

### Cell culture and pharmacological activation

P0 mouse (C57BL6J male and female pups) cortical neurons were plated (300,000 cells/well) and allowed to grow on glia cells. Plates were stored in an incubator (37°C) for the duration of the experiment. At DIV3, cells were infected with the viral mix of c-Fos-ZsGreen1-DR and CB7-mCherry. At DIV14, neurons were switched to an activating medium containing 4-AP (100 μM) and bicuculline (10 μM) for 30 min at 37°C. Then this activating medium was washed and TTX was added (0.5 μM) to prevent AP firing and stop neuronal activation. Neurons were then fixed using 4% paraformaldehyde (PFA) in 0.1 mM PBS at the different intervals following pharmacological activation. To calculate FLEN/c-Fos overlap, neurons were fixed as described above 3 hr after TTX was added to the medium.

### c-Fos immunofluorescence

Fixed cells were incubated with 50 mM ammonium chloride ($NH_4Cl$) in PBS for 10 min under constant shaking (1 ml/well) and washed three times with PBS (5 min each). Permeabilization was performed with 0.2% Triton X-100 in PBS for 5 min under shaking (1 ml/well), followed by three PBS washes (5 min each). Non-specific binding sites were blocked with 2% bovine serum albumin (BSA) in PBS for 60 min under shaking (1 ml/well). Primary antibody incubation was performed at RT for 60 min under shaking using rat anti-c-Fos (Synaptic Systems, ref 226 017) diluted 1:1000 in 2% BSA (160 μl/coverslip). Coverslips were washed three times with 2% BSA in PBS (5 min each) and incubated again in 2% BSA for 60 min under shaking (1 ml/well). Secondary antibody (goat anti-rat Alexa Fluor 647, Thermo Fisher Scientific, ref A21247) was applied at 1:500 dilution (160 μl/coverslip) for 60 min at RT under shaking and protected from light. Coverslips were washed three times in PBS (5 min each) and mounted using Fluoromount-G with DAPI (Southern Biotech).

### Cell counting

Culture and ex vivo slice images were processed using custom code. Each fluorescence channel (mCherry, ZsGreen, c-Fos, and DAPI) was normalized, Gaussian-filtered, and manually thresholded. Binary masks were generated and cleaned by removing small objects and segmented using watershed.

Masks were further filtered by area and shape (circularity, axis ratio, eccentricity, and solidity). From these counts, labeling ratios (between ZsGreen, mCherry, c-Fos, and DAPI) were calculated.

## Stereotaxic injection

Two-month-old C57BL6J mice were bilaterally injected with either AAV9-c-Fos-ZsGreen1-DR or the AAV9-RAM virus mixed with the infection control virus AAV9-CB7-mCherry into dorsal hippocampus. Mice were anesthetized with 4% isoflurane and placed in a stereotaxic apparatus, where they were kept under 1.5–2% isoflurane anesthesia (0.8–1 l/min), and injected with buprenorphine (0.1 mg/kg) and carprofen (5 mg/kg) to prevent post-surgery pain. The injection volume and flow rate (300 nl at 60 nl/min) were controlled with an injection micro-pump (World Precision Instruments). Injection coordinates, using bregma as a reference point, were as follows: AP –1.78, ML ±2.40, DV –2.35. This site targets dorsal CA3b, but virus spreading resulted in the infection of CA3a, CA3b, CA3c, CA2, and in some cases, few granule neurons and hilar cells of the DG. Mice were carefully monitored for 4 days following surgery, and behavioral manipulation started after a recovery period of approximately 2 weeks.

## Behavioral tests

Once recovered, mice were single-housed to reduce uncontrolled experiences that could contaminate the ensemble of CFC-related labeled engram cells. Then, mice were handled to habituate to the experimenter for at least 3 consecutive days prior to CFC. Mice injected with the RAM virus were deprived of doxycycline, switching to a regular diet, 48 hr before CFC training. On the one-trial learning day, mice were transported in the experimental room and immediately placed in a novel conditioning chamber enclosed in a sound-attenuating cabinet (Ugo Basile), enriched with visual cues, and allowed to explore it for a total duration of 300 s. After an initial exploration period, three mild foot-shocks (2 s, 0.7 mA) were delivered at 150, 210, and 270 s, leaving 30 s for additional context sampling at the end. Mice were recorded and freezing behavior was scored using ANY-Maze 6 software (Stoelting Europe). Freezing was detected as the absence of movement apart from respiration, and freezing events were scored when immobility lasted more than a threshold of 300 ms. Mice were then returned to their home cage. For context-only (CO) exploration, mice were placed in a 50 cm × 50 cm custom-built open field, where some objects were distributed. Mice were allowed to explore for 5 min before returning to their home cage. The immediate shock test was carried out in the fear conditioning system. Mice explored the conditioning chamber for only 2 s before receiving a 6-s-long foot-shock and then returned to their home cage.

## Histology

Mice were anesthetized with intraperitoneal administration of a ketamine/xylazine mix (100/10 mg/kg) at different time intervals following CFC (1.5, 3, 4.5, 6, and 12 hr for c-Fos-ZsGreen1-DR, 24 hr for RAM). Mice were then intracardially perfused with 0.9% NaCl solution followed by 4% PFA in 0.1 mM PBS. Brains were removed and postfixed in 4% PFA overnight at 4°C. Brains were transferred to sucrose 30% in 0.1 mM PBS at 4°C until they sank. Brains were cut on a cryostat into 25-µm-thick coronal sections and collected in PBS with 0.3% Triton X-100. Slices were immediately mounted on glass slides and covered in DAPI-based mounting medium. The number of fluorescent neurons was assessed directly without amplification.

## Confocal microscope

Confocal acquisitions were performed using a Leica SP8 White Light Laser 2 on an inverted stand (Leica Microsystems, Mannheim, Germany) and with a x20 (NA 0.75) immersion objective. Images were analyzed using ImageJ. At least 5 slices per animal were acquired. Imaging of neuronal cultures was carried out as well using confocal Leica SP8 White Light Laser 2 (Leica Microsystems, Mannheim, Germany) and with a x20 (NA 0.75) immersion objective. In all experiments, the settings (laser power, gain, offset) were fixed between groups as well as the pixel size (120 nm).

## Ex vivo slice preparation

Slices were prepared 3 or 24 hr following CFC for c-Fos-ZsGreen1-DR-injected mice and RAM-injected mice, respectively. To attempt to reduce stress-related unspecific c-Fos activation, mice were

anesthetized with a ketamine/xylazine mix (100/10 mg/kg; i.p.) and intracardially perfused with an ice-cold oxygenated (95% $O_2$ and 5% $CO_2$) cutting solution containing (in mM): 87 NaCl, 2.5 KCl, 1.25 $NaH_2PO_4$, 25 $NaHCO_3$, 25 glucose, 75 sucrose, 0.5 $CaCl_2$, 7 $MgCl_2$, pH 7.4, 315 mOsm/kg. Brains were quickly dissected out and parasagittal slices from both hemispheres (300 µm thick) were cut on a vibratome (Leica VT1200S, Germany) in the same oxygenated ice-cold cutting solution. Slices were then incubated at 34°C in a resting chamber containing the same cutting solution for 20–30 min for recovery. Slices were then transferred in a resting chamber filled with an oxygenated (95% $O_2$ and 5% $CO_2$) artificial cerebrospinal fluid (aCSF) containing (mM): 125 NaCl, 2.5 KCl, 1.25 $NaH_2PO_4$, 26 $NaHCO_3$, 1.3 $MgCl_2$, 2.3 $CaCl_2$, 10 glucose, pH 7.4, 300 mOsm/kg at room temperature for the rest of the day.

## Electrophysiology

When transferred to a recording chamber, slices were continuously superfused with aCSF. CA3 pyramidal cells were identified with an infrared differential interference contrast microscope with a water immersion 63X objective (N.A. 0.8). Neurons were then classified as engram neurons (simultaneous expression of ZsGreen1-DR/mCherry or EGFP/mCherry when RAM was used) or non-engram neuron (mCherry expression only) with a 2p microscope. Whole-cell recordings were made at room temperature using borosilicate glass capillaries with resistances between 4 and 6 MΩ. For current clamp recordings, the intracellular solution contained (in mM): 135 K-gluconate, 5 KCl, 2 NaCl, 10 $Na_2$-phosphocreatine, 0.1 EGTA, 10 HEPES, 5 Mg-ATP, 0.4 Na-GTP, pH 7.2 adjusted with KOH, 280–290 mOsm/kg. For voltage clamp mode, the patch pipettes were filled with a solution containing (mM): 140 Cs-methanesulfonate, 2 $MgCl_2$, 4 NaCl, 5 Na-phosphocreatine, 0.2 EGTA, 10 HEPES, 3 $Na_2$-ATP, 0.3 GTP, pH 7.2 adjusted with CsOH, 280–290 mOsm/kg. To record inhibitory currents, a high-chloride intracellular solution was used, containing (mM): 103 CsCl, 12 $CsCH_3O_3S$, 5 TEA-Cl, 10 HEPES, 4 Mg-ATP, 0.5 Na-GTP, 1 $MgCl_2$, pH adjusted to 7.2 with CsOH, 280–290 mOsm/kg. No liquid-junction correction was used. Neurons were held at –70 mV. Leak current and series resistance were monitored throughout each experiment; neurons with a leak current >200 pA and a series resistance >25 MΩ or if these parameters changed more than 20% were excluded from analysis. All recordings were performed 5–10 min after opening the membrane. Input resistance was quantified by linearly fitting the voltage change against injected current. Steps of hyper- and depolarizing current steps were injected to determine AP threshold and rheobase. AP firing pattern was determined by injecting 200 pA steps of current. sEPSCs were isolated by bath application of bicuculline (10 µM), while mEPSCs were recorded in the presence of bicuculline (10 µM) and TTX (0.5 µM). Spontaneous IPSCs were recorded in the presence of D-AP5 (50 µM) and NBQX (20 µM), while mIPSCs were isolated in the presence of D-AP5 (50 µM), NBQX (20 µM), and TTX (0.5 µM). Spontaneous and miniature synaptic events were detected using a deconvolution-based approach (*Pernía-Andrade et al., 2012*). To record evoked EPSCs, stimulating glass pipettes (World Precision Instruments) were filled with aCSF and placed in the hilus, close to the granule cell layer to stimulate initial portion of Mf axons. Voltage pulses (200 µs) were delivered through a stimulus isolator (Digitimer, UK). Stimulation intensity was adjusted to obtain minimal Mf stimulation (*Marchal and Mulle, 2004*). Basal synaptic transmission was recorded while holding CA3 neurons at –70 mV with pulses delivered at a low frequency of 0.1 Hz. Frequency was then increased to 1 Hz to induce frequency facilitation. After 2–3 min, when facilitation was extinguished, the voltage was switched to +5 mV (reversal potential for cations) to record di-synaptic eIPSC. All recordings were obtained with a HEKA EPC10 amplifier, filtered at 3.3 kHz and digitized at 10 kHz via PatchMaster software (Lambrecht, Germany). Data was analyzed offline using custom-made Python scripts.

## Statistical analysis

Analysis, statistics, and graphical representation were performed with custom-written Python codes using standard libraries (e.g. Numpy, Scipy, Pandas, and Matplotlib). All experiments consist of biological replicates, with *N* values reported throughout the manuscript. Data is shown as box plots and scatter plots. Scatter plots indicate individual samples per group. Boxplots indicate median, first and third quartiles with whiskers representing the rest of the distribution. Bold horizontal and vertical lines on scattered data points represent mean and SEM, respectively. All values were first tested for normality (D'Agostino and Pearson omnibus test or Shapiro–Wilk test). Normally distributed data were compared using a *t*-test (two-sided) while Mann–Whitney rank test (two-sided) was used for

non-normal dataset. Multiple group comparison was carried out using one-way ANOVA when data were normally distributed or Kruskal–Wallis when normality did not apply. Tukey's post hoc comparison analysis or Dunn post hoc comparison analysis was applied. Cumulative frequency distributions and normalized distributions were tested using the Kolmogorov–Smirnov test. Statistical differences were considered as significant at $p < 0.05$. In figure panels, statistical signficance is indicated as $*p<0.05$; $** p<0.01$; $***p<0.001$.

## Acknowledgements

We are grateful to Ashley L Kees, Ruth Betterton, Gäel Barthet, and Christine E Gee for helpful discussion. This work was supported by the Centre National de la Recherche Scientifique, the European Union Horizon 2020 European Training Network under Marie Skłodowska-Curie Action (SyDAD, Grant Agreement No. 676144 to CM and DC), the Fondation pour la Recherche Médicale (FDT201904007898 to DC) and LabEX Brain (to CM).

## Additional information

### Funding

| Funder | Grant reference number | Author |
|---|---|---|
| Agence Nationale de la Recherche | Hippencode 14-CE13-0015 | Christophe Mulle |
| European Union MSCA ITN SYDAD | 676144 | Dario Cupolillo |
| Fondation pour la Recherche Médicale | FDT201904007898 | Dario Cupolillo |

The funders had no role in study design, data collection, and interpretation, or the decision to submit the work for publication.

### Author contributions

Dario Cupolillo, Conceptualization, Formal analysis, Investigation, Writing – original draft, Writing – review and editing; Noelle Grosjean, Data curation, Investigation, Methodology, Writing – review and editing; Catherine Marneffe, Julio Viotti, Celia Reynaud, Investigation; Severine Deforges, Investigation, Methodology, Writing – review and editing; Mario Carta, Writing – original draft, Writing – review and editing; Christophe Mulle, Conceptualization, Supervision, Funding acquisition, Writing – original draft, Project administration, Writing – review and editing

### Author ORCIDs

Christophe Mulle (iD) https://orcid.org/0000-0003-2709-6615

### Ethics

C57BL/6J female mice were obtained from Janvier and cared for according to the regulations of the University of Bordeaux/CNRS Animal Care and Use Committee (agreement number, APAFIS#8444–-2016051911442448).

Reviewer #1 (Public review): https://doi.org/10.7554/eLife.105452.3.sa1
Reviewer #2 (Public review): https://doi.org/10.7554/eLife.105452.3.sa2
Author response https://doi.org/10.7554/eLife.105452.3.sa3

## Additional files

### Supplementary files
MDAR checklist

## Data availability

All data generated or analyzed in this study are included in the manuscript. Data are available online at Recherche Data Gouv (https://doi.org/10.57745/3U2NO5). Analysis codes used to process, analyze, and generate results from data are available online (https://github.com/dcupolillo/cupolillo_et_al_2025, copy archived at *Cupolillo, 2025*).

The following dataset was generated:

| Author(s) | Year | Dataset title | Dataset URL | Database and Identifier |
|---|---|---|---|---|
| Cupolillo D, Grosjean N, Marneffe C, Viotti J, Reynaud C, Deforges S, Carta M, Mulle C | 2025 | Early changes in the properties of CA3 engram cells explored with a novel viral tool in mice | https://doi.org/10.57745/3U2NO5 | Recherche Data Gouv, 10.57745/3U2NO5 |

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
