## [Editor Report · eLife Assessment]

This **important** study characterizes and validates a new activity marker - fast labelling of engram neurons (FLEN) - which is transiently active and driven by cFos, allowing the monitoring of intrinsic and synaptic properties of engram neurons shortly after the learning experience. The results **convincingly** demonstrate the utility of this novel viral tool for studying early changes in the properties of engram cells. FLEN will provide a beneficial tool for the neuroscience community once it is made available at a plasmid repository.

---

## [Referee Report · Reviewer #1 (Public review)]

Summary:

The manuscript by Cupollilo et al describes the development, characterization and application of a novel activity labeling system; fast labelling of engram neurons (FLEN). Several such systems already exist but this study adds additional capability by leveraging an activity marker that is destabilized (and thus temporally active) as well as being driven by the full-length promoter of cFos. The authors demonstrate the activity dependent induction and timecourse of expression, first in cultured neurons and then in vivo in hippocampal CA3 neurons after one trial contextual fear conditioning. In a series of ex vivo experiments the authors perform patch clamp analysis of labeled neurons to determine if these putative engram neurons differ from non-labelled neurons using both the FLEN system as well as the previously characterized RAM system. Interestingly the early labelled neurons at 3 h post CFC (FLEN+) demonstrated no differences in excitability whereas the RAM labeled neurons at 24h after CFC had increased excitability. Examination of synaptic properties demonstrated an increase in sEPCS and mEPSC frequencies as well as those for sIPSCs and mIPSCs which was not due to a change in the mossy fiber input to these neurons.

Strengths:

Overall the data is of high quality and the study introduces a new tool while also reassessing some principles of circuit plasticity in the CA3 that have been the focus of prior studies.

Weaknesses:

No major weaknesses were noted

---

## [Referee Report · Reviewer #2 (Public review)]

Summary:

Cupollilo et al. investigate the properties of hippocampal CA3 neurons that express the immediate early gene cFos in response to a single foot shock. They compare ex-vivo the electrophysiological properties of these "engram neurons" labeled with two different cFos promoter-driven green markers: Their new virally delivered tool FLEN labels neurons 2-6 h after activity, while RAM contains additional enhancers and peaks considerably later (>24 h). Since the fraction of labeled CA3 cells is comparable with both constructs, it is assumed (but not tested) that they label the same population of activated neurons at different time points. Both FLEN+ and RAM+ neurons in CA3 receive more synaptic inputs compared to non-expressing control neurons, which could be a causal factor for cFos activation, or a very early consequence thereof. Frequency facilitation and E/I ratio of mossy fiber inputs were also tested, but are not different in both cFos+ groups of neurons. One day after foot shock, RAM+ neurons are more excitable than RAM- neurons, suggesting a slow increase in excitability as a major consequence of cFos activation.

Strengths:

The study is conducted to high standards and contributes significantly to our understanding of memory formation and consolidation in the hippocampus. Modifications of intrinsic neuronal properties seem to be more salient than overall changes in the total number of (excitatory and inhibitory) inputs, although a switch in the source of the synaptic inputs would not have been detected by the methods employed in this study

Weaknesses:

The new tool FLEN is not quantitatively compared to e.g. the TetTag reporter mouse. Nevertheless, the fluorescent images of FLEN+ neurons are quite convincing.

---

## [Author Response]

The following is the authors’ response to the original reviews.

**Reviewer #1 (Public review):**
Summary:The manuscript by Cupollilo et al describes the development, characterization, and application of a novel activity labeling system; fast labelling of engram neurons (FLEN). Several such systems already exist but this study adds additional capability by leveraging an activity marker that is destabilized (and thus temporally active) as well as being driven by the full-length promoter of cFos. The authors demonstrate the activity-dependent induction and time course of expression, first in cultured neurons and then in vivo in hippocampal CA3 neurons after one trial of contextual fear conditioning. In a series of ex vivo experiments, the authors perform patch clamp analysis of labeled neurons to determine if these putative engram neurons differ from non-labelled neurons using both the FLEN system as well as the previously characterized RAM system. Interestingly the early labelled neurons at 3 h post CFC (FLEN+) demonstrated no differences in excitability whereas the RAMlabelled neurons at 24h after CFC had increased excitability. Examination of synaptic properties demonstrated an increase in sEPCS and mEPSC frequencies as well as those for sIPSCs and mIPSCs which was not due to a change in the mossy fiber input to these neurons.Strengths:Overall the data is of high quality and the study introduces a new tool while also reassessing some principles of circuit plasticity in the CA3 that have been the focus of prior studies.Weaknesses:No major weaknesses were noted.
**Reviewer #2 (Public review):**
Summary:Cupollilo et al. investigate the properties of hippocampal CA3 neurons that express the immediate early gene cFos in response to a single foot shock. They compare ex-vivo the electrophysiological properties of these "engram neurons" labeled with two different cFos promoter-driven green markers: Their new tool FLEN labels neurons 2-6 h after activity, while RAM contains additional enhancers and peaks considerably later (>24 h). Since the fraction of labeled CA3 cells is comparable with both constructs, it is assumed (but not tested) that they label the same population of activated neurons at different time points. Both FLEN+ and RAM+ neurons in CA3 receive more synaptic inputs compared to non-expressing control neurons, which could be a causal factor for cFos activation, or a very early consequence thereof. Frequency facilitation and E/I ratio of mossy fiber inputs were also tested, but are not different in both cFos+ groups of neurons. One day after foot shock, RAM+ neurons are more excitable than RAM- neurons, suggesting a slow increase in excitability as a major consequence of cFos activation.Strengths:The study is conducted to high standards and contributes significantly to our understanding of memory formation and consolidation in the hippocampus. Modifications of intrinsic neuronal properties seem to be more salient than overall changes in the total number of (excitatory and inhibitory) inputs, although a switch in the source of the synaptic inputs would not have been detected by the methods employed in this studyWeaknesses:With regard to the new viral tool, a direct comparison between the new tool FLEN and existing cFos reporters is missing.
**Reviewer #1 (Recommendations for the authors):**
I have only minor suggestions for the authors to consider.(1) In the in vitro characterization, the percentage of labelled neurons seems very low after a powerful and prolonged activation. It was somewhat surprising and raised the question of how accurately the FLEN construct reflects endogenous cFOS activity. Could the authors speak to this?

The reviewer is correct that the level of FLEN positive neurons, as compared to mCherry positive neurons, is low as compared to studies using viral infection with RAM vectors in neuronal cultures (Sorensen et al, 2016, Sun et al, 2020), which is around 70-80% following chemical stimulation. The authors do not provide evidence however for a comparison with endogenous c-Fos activity in cell cultures. The reason for a discrepancy in the effect of chemical stimulation of cultured neurons is not clear, but may depend on culture conditions which may vary between labs.

FLEN was constructed using a mouse c-Fos promoter (-355 to +109) (Cen et al, 2003). To answer the reviewer’s question we performed an additional experiment in cultured neurons in which we found that 77.1 % of FLEN positive neurons were also c-fos positive neurons (using immunocytochemistry).

(2) The authors compare the two labelling strategies and interpret their data with the presumption that both label a similar set of active neurons. This is particularly relevant when they suggest there might be a progressive increase in the excitability of active neurons with time. This is certainly a possibility, but the authors should also consider other possibilities that the two markers might label different populations of neurons. For example, if they require different thresholds for activation, it is possible that one is more sensitive to activity than the other. As these are unknown variables the authors should temper the interpretation accordingly.

Indeed, the reviewer is correct that this limitation should be discussed. We have added this as a point of discussion in the text (line 355-358). In the article describing the RAM strategy (Sorensen et al, 2016) the authors use RAM to label DG neurons activated during an experience in a context A (Figure 4). Exploiting the fact that engram cells are re-activated when the animal is re-exposed to the same environment of training (memory recall), they performed c-Fos staining 90 minutes following either context A or context B re-exposure. The RAM-c-Fos overlap percentage was higher in A-A rather than A-B (A-A was a bit more than 20%). This means that RAM has captured a group of cells during training that, at least in part, were re-activated during recall. This could in part support the assumption that RAM and c-Fos share a certain overlap. Of course, this was done in DG, while we worked in CA3. In addition, both strategies label in their great majority c-Fos+ neurons (see above answer to point #1). This can not completely rule out the possibility that FLEN and RAM label partly distinct population of activated cells.

(3) An increase in the frequency of synaptic events is observed in neurons labelled with both markers. The authors propose that this may be due to an increase in synaptic contacts based on prior studies. However, as this is the first functional assessment why not consider changes in release probability as a mechanism for this finding?

We have added this as a possibility in the text (line 362-363).

(4) It would be useful to include plots of the average frequency of m/sEPSCs and m/sIPSCs in Figures 4 and 5. These figures could also be combined into a single figure.

We agree with the reviewer that figure 4 and 5 could be merged into a single figure. In the revised version, figure 5A becomes panel C in figure 4. Text and figure descriptions were adjusted accordingly.

**Reviewer #2 (Recommendations for the authors):**
(1) Abstract, line 24: "In contrast, FLEN+ CA3 neurons show an increased number of excitatory inputs." RAM+ neurons also show an increased number of excitatory inputs, so this is not "in contrast". Also, not just excitatory, but also inhibitory synaptic inputs are more numerous in cFos+ neurons. Please improve the summary of your findings.

“In contrast” referred to the fact that FLEN+ neurons do not show differences in excitability as compared to FLEN- neurons, as mentioned in the previous sentence. We now provide a more explicit sentence to explain this point: “On the other hand, like RAM+ neurons, FLEN+ CA3 neurons show an increased number of excitatory inputs.”

(2) Novel tool: Destabilized cFos reporters were introduced 23 years ago and are also part of the TetTag mouse. I am not sure that changing the green fluorescent protein to a different version merits a new acronym (FLEN). To convince the readers that this is more than a branding exercise, the authors should compare the properties (brightness, folding time, stability) of FLEN to e.g. the d2EGFP reporter introduced by Bi et al. 2002 (J Biotechnol. 93(3):231) and show significant improvements.

We thank the reviewer for this comment which compelled us to evaluate the features of other tools used to label neurons activated following contextual fear conditioing. The key properties of FLEN as compared to other tools used to label engrams is that: (i) it is a viral tool, as opposed to transgenic mice, (ii) a c-fos promoter drives the expression of a brightly fluorescent protein allowing their identification ex vivo for functional analysis, (iii) the fluorescent protein is rapidly destabilized, providing the possibility to label neurons only a few hours after their activation by a behavioural task.

We did not find any viral tools providing the possibility to label c-fos activated neurons for functional assesment. We have not been able to find references for the use of the d2EGFP reporter introduced by Bi et al. 2002 in a behavioural context. One of the major difference and improvement is certainly the brightness of ZsGreen. In cell cultures, ZsGreen1 showed a 8.6-fold increase in fluorescence intensity as compared with EGFP (Bell et al, 2007).

Amongst tools with comparable properties, eSARE was developed based on a synthetic Arc promoter driving the expression of a destabilized GFP (dEGFP) (Kawashima et al 2013). We initially used ESARE–dGFP but unfortunately, in our experimental conditions we found that the signal to noise ratio was not satisfactory (number of cells label in the home cage vs. following contextual fear conditining).

We developed a viral tool to avoid the use of transgenic reporter lines which require laborious breeding and is experimentally less flexible. Nevertheless, many transgenic mice based on the expression of fluorescent proteins under the control of IEG promoters have been developed and used. Some of these mice show a time course of expression of the transgene which is comparable to FLEN. For instance, in organotypic slices from Tet-Tag mice, the time course of expression of EGFP slices follows with a small delay endogenous cFOS expression, and starts decaying after 4 hours (Lamothe-Molina et al, 2022). However, the fluorescence was too weak to visualize neurons in the slice (Christine Gee, personal communication), and imaging is perfomed after immunocytochemistry against GFP.

Therefore, we feel that the name given to the FLEN strategy is legitimate. The features of the FLEN strategy were summarized in the discussion (Lines 318-322).

(3) Line 214: "...FLEN+ CA3 PNs do not show differences in [...] patterns of bursting activity as compared to control neurons." It looks quite different to me (Figure 3E). Just because low n precludes meaningful statistical analysis, I would not conclude there is no difference.

We agree with the reviewer that the data in Figure 3E are not conclusive due to small sample size, which limits the reliability of statistical comparison. Additionally, the classification of bursting neurons is highly dependent on the specific criteria used, which vary considerably across the literature. To avoid overinterpretation or misleading conclusions, we decided to remove the panel E of Figure 3 showing the fraction of bursting neurons. Nevertheless, we draw the attention to the more robust and interpretable results: RAM⁺ neurons exhibit an increase in firing frequency and a distinct action potential discharge pattern, data which we believe are informative of altered excitability.

(4) Line 304: Remove the time stamp.

This was done.

(5) Line 334: "...results may be explained by an overall increased activity of CA1 neurons..." I don't understand - isn't CA1 downstream of CA3?

The reviewer is correct that the sentence was misleading. We removed the reference to CA1, as it was more of a general principle about neuronal activity.

(6) Line 381: "resolutive", better use "sensitive".

This was changed.

(7) Figure S3: Fear-conditioned animals were 3 days off Dox, controls only 2 days. As RAM expression accumulates over time off Dox, this is not a fair comparison.

We thank the reviewer for pointing out the incorrect reporting of the experimental design in Figure S3 panel A (bottom), which could lead to misinterpretation of results. In fact, the two groups of mice (CFC vs. HC) underwent all experimental steps in parallel. Specifically, both groups were maintained on and off Doxycycline for the same duration and received viral injection on the same day. 48 hours after Dox withdrawal, the CFC group was trained for contextual conditioning, while the HC group remained in the home cage in the holding room. All animals were thus sacrificed 72 hours after Dox removal. We have corrected the figure to accurately reflect this timeline.

(8) Please provide sequence information for c-cFos-ZsGreen1-DR. Which regulatory elements of the cFos promoter are included, is the 5' NTR included? This information is very important.

The information is now provided in the Methods section.

(9) Please provide the temperature during pharmacological treatments (TTX etc.) before fixation.

The pharmacological treatment was performed in the incubator at 37°C, this is now indicated in the methods.